# Research on the Dynamic Response of a Continuous Steel Box Girder Bridge Based on the ANSYS Platform

Baitian Wang [1,2,*] , Yudong Jia [3], Hongjuan Zhao [2], Simeng Wang [2], Zhengchuang Jin [2] and Jinfeng Yang [2]

1 School of Civil Engineering, Nanjing Tech University, Nanjing 210000, China
2 School of Architecture and Civil Engineering, Shangqiu University, Shangqiu 476113, China
3 China Construction Eighth Engineering Bureau Second Construction Co., Ltd., Jinan 250100, China
* Correspondence: 202162126003@njtech.edu.cn

**Abstract:** Under the action of various dynamic loads, bridges will experience large deflections and stress. When the situation is difficult, it will affect the regular use of the bridge and even cause it to collapse suddenly. This article generated a sample of road surface irregularities based on the Chinese national standard. An ANSYS model was used to create the vehicle–bridge coupling model. In order to meet the actual engineering calculations, an essential but valuable analytical approach is presented here. The node coupling method established the time-varying vehicle axle coupling system. The moving tire force was applied to the axle coupling system. The ANSYS parametric design language was adopted to realize the process of the vehicle approach and exit of the bridge. Combined with the actual data of dynamic and static load experiments, the model's accuracy was verified. The influence of different vehicle driving speeds, road irregularities, vehicle driving position, and vehicle driving state are analyzed in this paper. The vehicle speed had no significant influence on the displacement time-history and the force of the middle wheel of the vehicle at a specific driving position. The pavement grade significantly influenced the bridge's displacement time-history and acceleration spectrum.

**Keywords:** bridge coupling effect; continuous steel box girder bridge; road roughness; dynamic response; node coupling method

## 1. Introduction

In the last century, the bridge vibration phenomenon caused by mobile vehicles has been the focus of relevant researchers. The moving force exerted on the bridge is mainly the load that changes with time and force position. Since the middle of the 19th century, these forces and effects have been widely discussed.

Researchers have paid more attention to the relationship between railway bridges and trains, mainly because of the oversized load, small unit quality, and small bridge width. The coupling effect between train and bridge must be considered [1,2]. With the extensive use of lightweight continuous steel box girder bridges in urban bridges and the increase in urban vehicles year by year, the problems caused by the vehicle bridge coupling effect of this kind of bridge are becoming more prominent. Therefore, it is necessary to discuss the interaction between vehicles and bridges of this kind of continuous steel box girder bridge.

The axle coupling model established at the beginning of the last century has been relatively simplified, such as by Keylov and Timoshenko, who analyzed the generation mechanism of the resonance phenomenon between the bridge and the vehicle [3,4]. In the 1840s, Inglis considered the effect of the vehicle and bridge's inertial force. The calculated solution was close to the actual measured data [5]. In 1956, Mise and Kunii supplemented and corrected the Inglis theory to establish the corresponding computational method [6]. In the 1960s, R.K.Wen discussed the problem of two-axle vehicles crossing a bridge [7]. Since the 1980s, domestic and foreign scientific research institutions have fully realized the

importance of vehicle bridge coupling vibration and began to invest a lot of energy and time in studying the vehicle bridge coupling dynamics caused by the irregularity of the bridge deck [8].

In recent years, scholars have performed detailed studies on the vehicle–bridge interaction model. Ge [9] developed a freezing technique to consider the vehicle axle coupling time-varying model. Stoura developed the MBS (modified bridge system) method to solve the control equations of bridges and vehicles. This method has a high accuracy and computational efficiency [10]. A time integral analysis method for the vehicle bridge interaction problem was proposed [11]. Xia proposed to treat the contact force as the bridge subsystem, the vehicle, and the bridge subsystem's external load and solve the two subsystems, respectively [12,13]. Seves et al., proposed methods that avoid using time-dependent system matrices [14]. Xiao proposed a stochastic analysis method based on GHW (Generalized Harmonic Wavelet) [15]. Shi and N.Uddin established multiple models of boundary conditions fixed at both ends, fixed superficial branches, and one end and free (cantilever). L.Ma selected 15 continuous girder bridges for dynamic amplification factor analysis. It was found that when the resonance phenomenon of continuous girder bridges occurs, the damping coefficient of continuous girder bridges increases significantly.

Scholars have also completed innovative research related to coupling loads. For example, Sham explained the seismic behavior of a new isolated monorail bridge [16]. Bucinskas introduced a vehicle simplified model, a wheel–rail nonlinear interaction model, a bridge structure finite element (FE) model, and a layered soil semi-analytical mode [17]. Camara considered the impact of oblique wind on driving safety [18].

With the application of powerful general FEM (finite element method) software, relevant researchers have paid more attention to the application of general FEM software in the calculation of vehicle and bridge motion and have tried to develop related algorithms for the interaction between vehicle and bridge. Researchers hoped to use the function of finite element software to solve the problem of the interaction between vehicles and bridges. Since ANSYS has perfect and unique functions in front and back processing, ANSYS can use the FORTRAN language to realize the transmission of vehicle and bridge interaction force, so ANSYS is suitable for compiling vehicle and bridge interaction programs.

To sum up, the calculation algorithm for the bridge interaction model is mainly realized in two ways: the coupling algorithm for the overall solution and the iterative algorithm for the separate solution. Scholars have put forward many calculation models, mainly for solving railway bridge problems, which have been applied less frequently to practical engineering and urban steel box girder bridges. In this paper, based on the dynamic and static load experiment of five continuous continuous-span span-steel steel-box box-girder bridges, the vehicle and bridge models were established, respectively, on the ANSYS platform. The system was formed based on the node coupling method. The vehicle bridge interaction force was decomposed correctly, avoiding the complex iterative process with a clear concept. In order to solve the difficulties of engineers in practical application, a simple, feasible and accurate analysis method is proposed in this paper. The influence of different vehicle speeds, road grade, vehicle driving lateral position, vehicle braking and jumping on the displacement time range, and the acceleration frequency domain of the bridge span structure are analyzed.

## 2. Road Surface Irregularities Sample

Road roughness mainly reflects the concavity and convexity of the road along the centerline. The Chinese national standard (GB/T7031) 'Mechanical vibration-Road surface profiles-Reporting of measured data' [19] adopts Formula (1) as the power spectrum function.

$$S_q(\Omega) = S_q(\Omega_0)(\Omega/\Omega_0)^{-\omega}, \Omega > 0 \tag{1}$$

In the formula, $\Omega$ is the spatial frequency (secondary/m); $\Omega_0$ is the spatial reference frequency (secondary/m), generally = 0.1 (secondary/m); $\omega$ is the frequency coefficient,

generally take 2. The broadband is $(\Omega_1, \Omega_2)$, $\Omega_1$ and $\Omega_2$ indicates the effective frequency band. The value can be determined by referring to the specifications.

After inverse Fourier transforms, the distribution function of bridge deck unevenness can be obtained by trigonometric series superposition according to Formula (2).

$$y(x) = \sqrt{2} \sum_{k=1}^{N} \sqrt{S_q(\Omega_k)\Delta\Omega} \cos(2\pi\Omega_k + \varphi_k) \tag{2}$$

$\Omega_1$, $\Omega_N$ in the formula, $y(x)$ is the generated unevenness sample sequence; $S_q(\Omega_k)$ is the power spectral density calculated according to Chinese national standards; $\Omega_k(k = 1, 2, \cdots, N)$ is the frequency of investigation; $\Omega_1$, $\Omega_N$ are reciprocal of the maximum wavelength and the reciprocal of the minimum wavelength; $\Delta\Omega$ is the step size of the frequency to be calculated; $\varphi_k$ is the phase of the frequency, which can be taken uniformly or randomly.

In the formula, the uneven sample sequence is the power spectral density calculated according to the Chinese national standard; $\Omega_k$ is the investigated frequency; the inverse of the maximum wavelength and the smallest wavelength; the step of the calculated frequency; the phase of the kth frequency, which can be taken evenly or at random.

Figure 1a–d shows the roughness of pavement grades A, B, C, and D, respectively, simulated according to national standards.

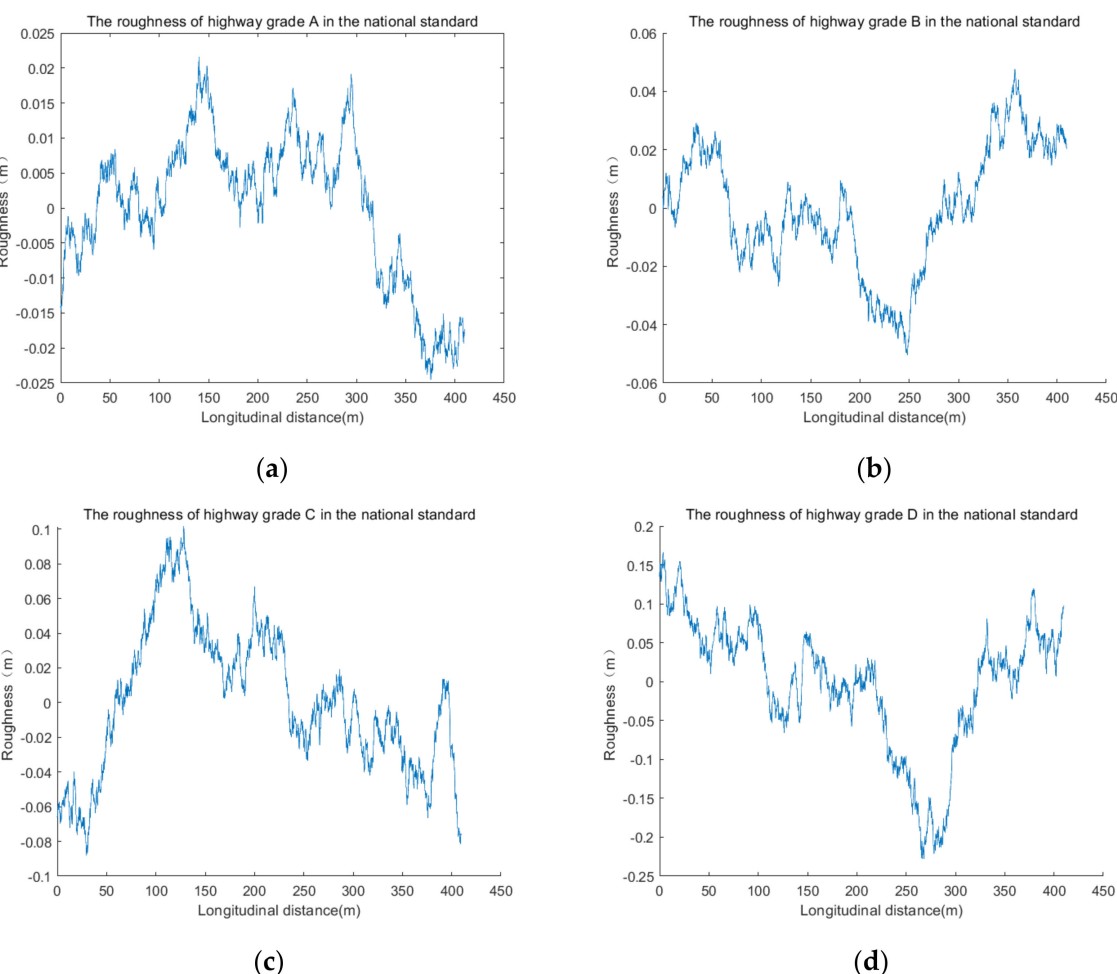

**Figure 1.** Roughness of different pavement grades (m). (**a**) The roughness of pavement grade A. (**b**) The roughness of pavement grade B. (**c**) The roughness of pavement grade C. (**d**) The roughness of pavement grade D.

## 3. Decomposition of Vehicle Force and Steps of the Vehicle Axle Coupling Calculation

### 3.1. Vehicle Force Decomposition

In order to facilitate the consideration road roughness effect in the ANSYS platform, it was assumed that the force on the bridge could be decomposed into the following three aspects:

(1) The vehicle force is caused by gravity, which is a constant moving force that changes with the vehicle's position. This force has nothing to do with the bridge vibration, which can be defined as the gravity effect of vehicle force;

(2) The vehicle force is caused by the road roughness. This force is the moving variable force that changes with the vehicle's position on the bridge. Because the road roughness is random, this variable force is random. Assuming that the force has nothing to do with the vibration of the bridge, it is advisable to define the variable force as the random effect of vehicle force. If the road surface is smooth, the force is zero. For the convenience of calculation, the force is considered as the external force;

(3) Applying the gravity effect of vehicle force and the random effect of vehicle force to the vehicle bridge coupling system will inevitably cause bridge vibration. The vibration of the bridge will cause vehicle vibration. Because the force generated by the coupling movement of the vehicle and the bridge (which may be defined as the coupling effect of vehicle force) is closely related to the bridge's vibration, it is necessary to solve the force in axle coupling system.

As shown in Figure 2, the steps of the coupling system were calculated. The node coupling method realized the establishment of a vehicle axle coupling system.

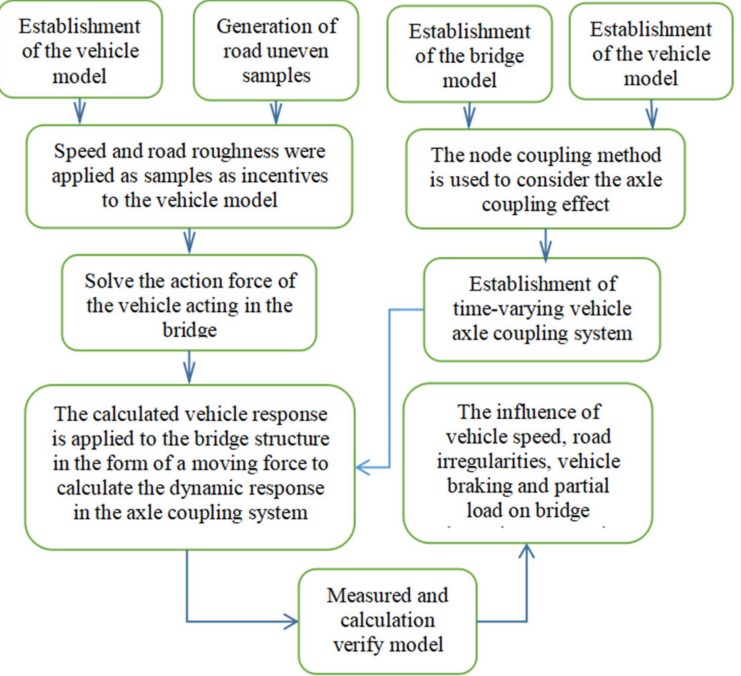

**Figure 2.** Procedures for the calculation of the vehicle axle coupling system.

### 3.2. Establishment of the Vehicle Model

Figure 3 shows the vehicle model. In this model, the vehicle body was rigid without deformation, the vehicle suspension and tire were spring dampers with energy dissipation, and the corresponding mass was assumed to be mass points. The relevant parameters of the vehicle were set in reference [20], as shown in Table 1 and Figure 3a. There were nine degrees of freedom in the model, including the up and down, the nodding motion, the rocking motion in the transverse bridge direction, and the up and down motion of six mass blocks. The other degrees of freedom were constrained or coupled with the bridge.

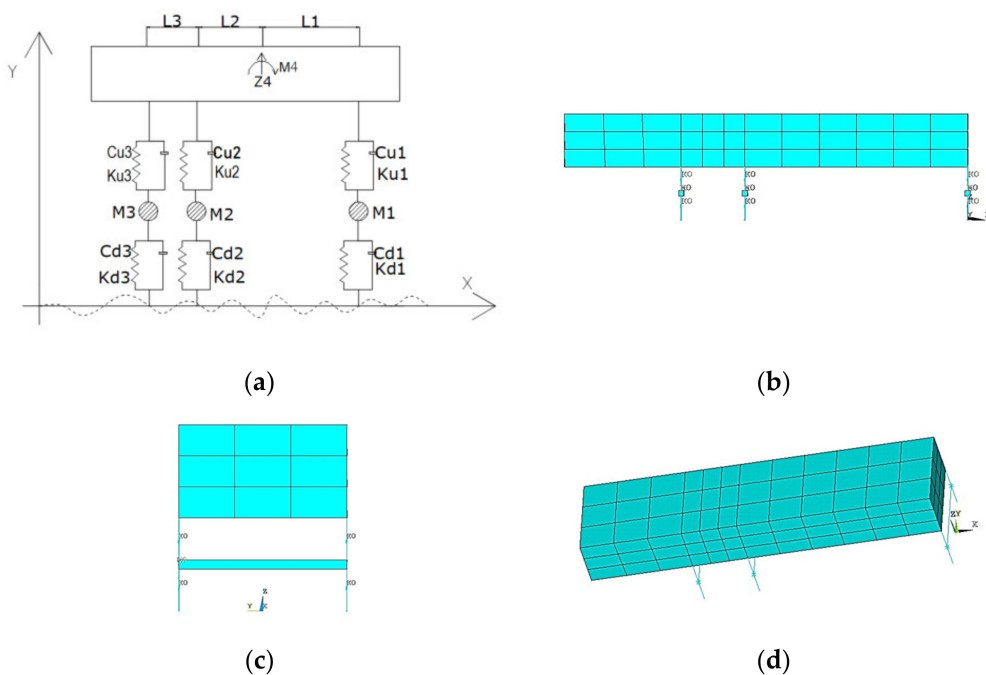

**Figure 3.** Vehicle model. (**a**) Model parameter; (**b**) Side view; (**c**) Face view; (**d**) Axonometric drawing.

**Table 1.** Vehicle model parameters and their representative meanings.

| Parameter Name | Physical Meanings | Numerical Size | Unit |
|---|---|---|---|
| $M_1$ | Front-wheel and suspension mass | 500 | kg |
| $M_3$ | Medium wheel and suspension mass | 1450 | kg |
| $M_3$ | Rear-wheel and suspension mass | 1450 | kg |
| $M_4$ | Body mass | 28,500 | kg |
| $K_{u1}$ | Front-wheel suspension elasticity coefficient | 1,577,000 | N/m |
| $K_{u2}$ | Medium-wheel suspension elasticity coefficient | 4,724,000 | N/m |
| $K_{u3}$ | Rear-wheel suspension elastic coefficient | 4,724,000 | N/m |
| $C_{u1}$ | Front-wheel suspension damping factor | 112,000 | kg/s |
| $C_{u2}$ | Mid-wheel suspension damping coefficient | 334,200 | kg/s |
| $C_{u3}$ | Rear-wheel suspension damping coefficient | 334,200 | kg/s |
| $K_{d1}$ | Front-wheel tire elasticity coefficient | 3,146,000 | N/m |
| $K_{d2}$ | Medium-wheel tire elasticity coefficient | 4,724,000 | N/m |
| $K_{d3}$ | Rear-wheel tire elasticity coefficient | 4,724,000 | N/m |
| $C_{d1}$ | Front-wheel tire damping coefficient | 13,300 | kg/s |
| $C_{d2}$ | Mid-wheel tire damping coefficient | 10,000 | kg/s |
| $C_{d3}$ | Rear-wheel tire damping coefficient | 10,000 | kg/s |
| $L_2$ | Distance between the front wheel and the center | 3.8 | m |
| $L_1$ | The distance between the middle wheel and the center | 0.4 | m |
| $L_3$ | Distance between the middle and rear wheels | 1.2 | m |
| $L_0$ | Left and right wheel distance of the vehicle | 1.8 | m |

When building the vehicle model on the ANSYS platform, the solid65 element was used to simulate the rigid vehicle body, the combin14 spring damping element was used to simulate the suspension and tire, and the mass21 element was used to simulate the mass effect of the suspension and the tires. Figure 3b–d shows the vehicle finite element model established on the ANSYS platform, which was a relatively simplified model.

Table 2 shows the natural vibration frequency of the vehicle. The calculated fundamental frequency was 2.49, which is consistent with the fundamental frequency of the actual vehicle. Figure 4 shows the vehicle mode diagrams of orders 1–2.

**Table 2.** Vehicle Modes.

| Mode Order | Modality | Vibration Characteristics |
|:---:|:---:|:---:|
| 1 | 2.49 | Car-body nodded |
| 2 | 2.77 | Car-body swing |
| 3 | 3.11 | Car-body twist |
| 4 | 11.05 | Suspension vibration |
| 5 | 11.06 | Suspension vibration |

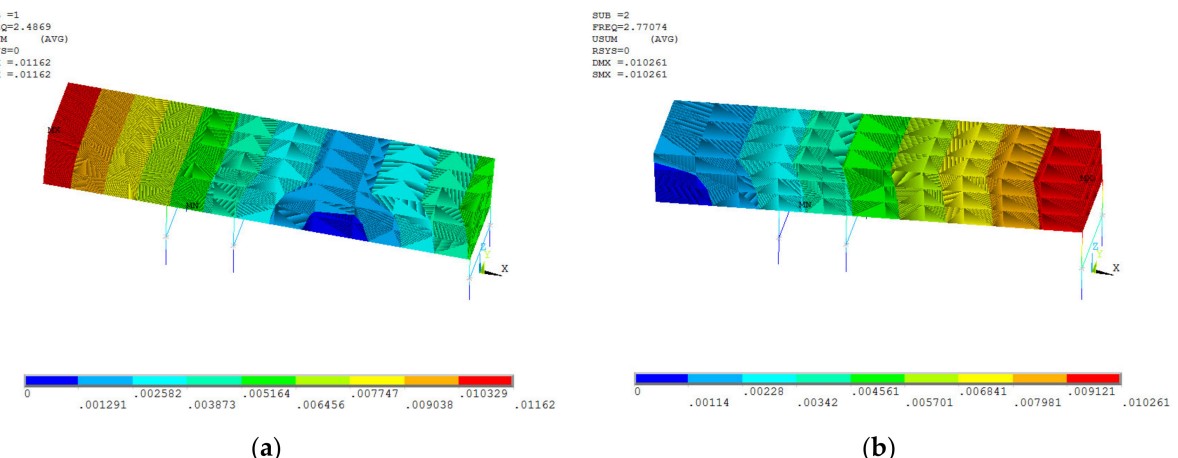

(**a**)                                           (**b**)

**Figure 4.** Order 1–2 self-vibration modes of vehicles. (**a**) Order 1; (**b**) Order 2.

*3.3. The Acting Force of the Vehicle Acting on the Bridge*

Vehicles driving on the bridge produce complex dynamic loads due to the influence of bridge deck roughness. When the vehicle crosses the bridge, the tire and road surface are close and appropriate, and it is assumed that there is no void phenomenon. The force on the wheel is the same as the force applied by the wheel. It is the relationship between the acting force and the reaction force. Using the condition that the force and displacement are consistent, the force value on the bridge can be derived.

As shown in Equation (3):

$$[M_v]\left\{\ddot{Z}_v\right\} + [C_v]\left\{\dot{Z}_v\right\} + [K_v]\{Z_v\} = \{P_v\} \tag{3}$$

where $[M_v], [C_v], [K_v]$ donate the vehicle mass matrix, damping matrix, stiffness matrix; $\{P_v\}$ is the overall external force vector, $\{Z_v\}$ the vehicle displacement column vector, and the acceleration vector; the above matrix is formed in the ANSYS platform.

Equation (4) shows the force equation based on the vehicle model.

$$F_i = C_{di}(\dot{\Delta}_i) + K_{di}(\Delta_i) \tag{4}$$

where, $\Delta_i$ is the relative value of the vertical displacement. $\Delta_i = Z_i - x_i - y_i$, $Z_i$ is the vertical displacement of the wheelset; $x_i$ is the vertical displacement of the bridge at the corresponding position of the action point of the wheelset. In this paper, the value of this item was 0 when calculating the force of vehicles on the bridge; $y_i$ is the roughness of the corresponding position.

## 4. Force Characteristics of Vehicle Tires

*4.1. Influence of Irregularities and Speed on the Vehicle Tire Force*

The tire force was extracted based on the ANSYS vehicle model and the road roughness. The force characteristics were analyzed. The corresponding speed of the vehicle was realized through different time steps.

　　　　Figure 5a,b shows the force curves of the middle wheel and tire of the vehicle with the driving time under different road grades. Figure 5c,d show the middle wheel's spectrum curve of tire force at different road grades. Figure 6 shows tire force's time-history and Fourier spectrum curves at different driving speeds. Figure 7a shows the variation law curve of the dynamic coefficient at different driving grades. Table 3 shows the peak force value under corresponding speed conditions under different road grades. In this paper, the value of the dynamic coefficient was the ratio of the maximum dynamic effect of the structure to the corresponding static effect.

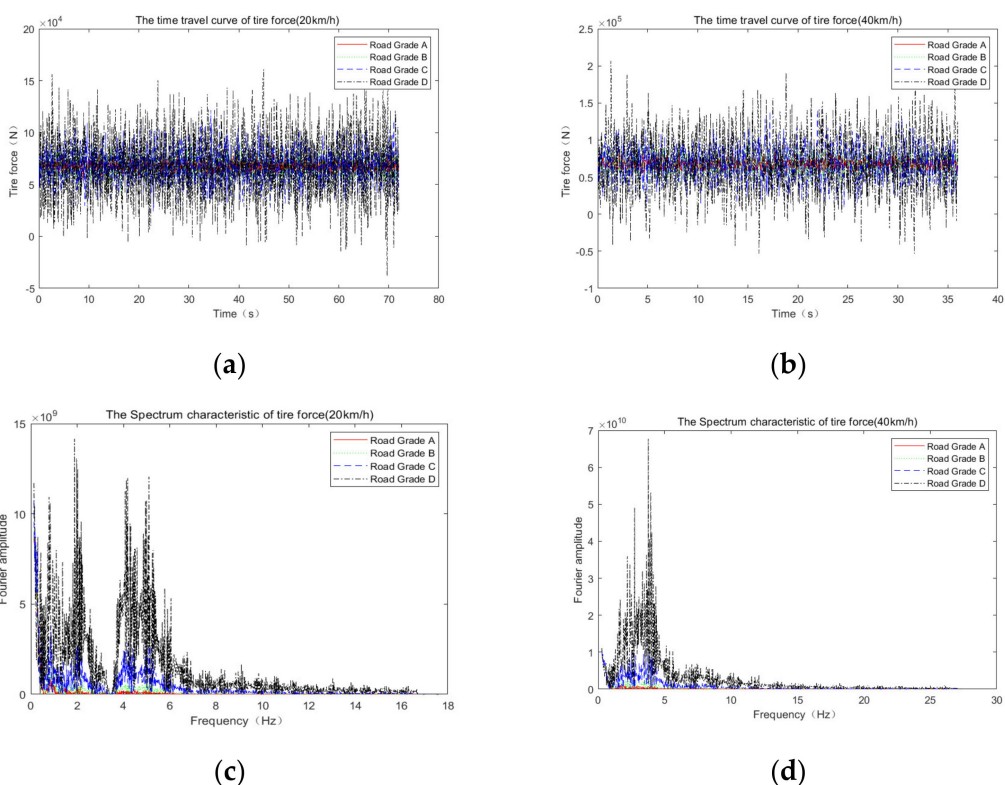

**Figure 5.** Time-history curves and Fourier spectra of tire force to different road grades. (**a**) Time-history curve for the case of speed 20 km/h; (**b**) Time-history curve for the case of speed 40 km/h; (**c**) Force action Fourier spectrum curve of speed 20 km/h; (**d**) Force action Fourier spectrum curve of speed 40 km/h.

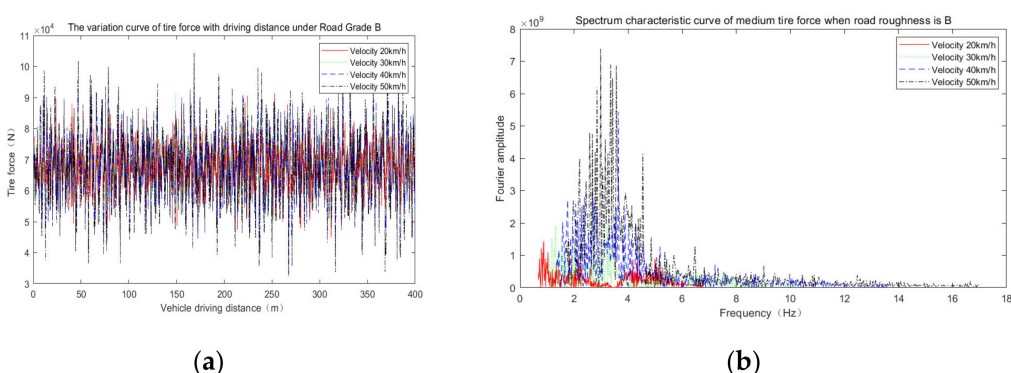

**Figure 6.** Time-history and characteristic spectrum curves of tire force at different driving speeds. (**a**) Time-history curve of tire force at a pavement grade B; (**b**) Tire force Fourier spectrum at the pavement grade B.

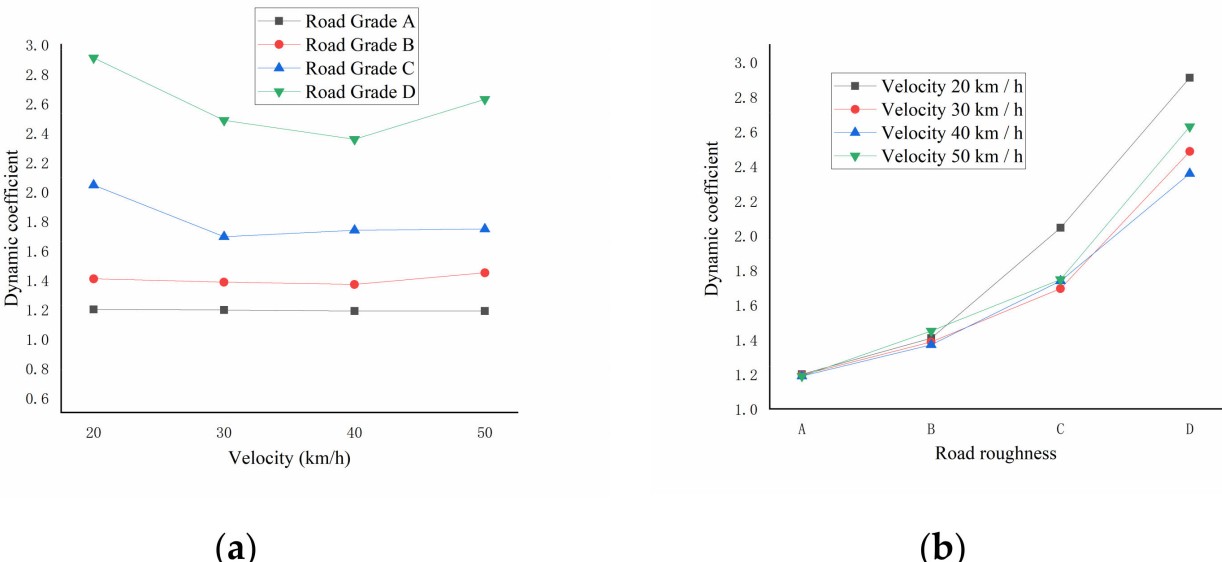

**Figure 7.** Peak wheel force in the vehicle. (**a**) The dynamic coefficient of the wheel at different driving grades; (**b**) the dynamic coefficient at different driving speeds.

**Table 3.** Peak wheel action force in vehicles.

| Road Grade | Vehicle Speed (km/h) | Middle-Wheel Acting Force (N) | Static Load (N) | Dynamic Coefficient |
|---|---|---|---|---|
| grade A | 20 | 84,643 | 70,525 | 1.200 |
| | 30 | 84,380 | 70,525 | 1.196 |
| | 40 | 83,880 | 70,525 | 1.189 |
| | 50 | 83,857 | 70,525 | 1.189 |
| grade B | 20 | 99,333 | 70,525 | 1.408 |
| | 30 | 97,724 | 70,525 | 1.386 |
| | 40 | 96,620 | 70,525 | 1.370 |
| | 50 | 102,211 | 70,525 | 1.449 |
| grade C | 20 | 144,205 | 70,525 | 2.045 |
| | 30 | 119,480 | 70,525 | 1.694 |
| | 40 | 122,578 | 70,525 | 1.738 |
| | 50 | 123,242 | 70,525 | 1.747 |
| grade D | 20 | 205,249 | 70,525 | 2.910 |
| | 30 | 175,250 | 70,525 | 2.485 |
| | 40 | 166,237 | 70,525 | 2.357 |
| | 50 | 185,361 | 70,525 | 2.628 |

It can be seen from Figure 5 that the road grade had a noticeable impact on a load of vehicles, and the peak force of vehicles gradually increased with pavement deterioration. The peak value of the spectrum curve mostly appeared at the frequency of 0.5–5.0 Hz, and the amplitude of the spectrum will increase with the deterioration of pavement. From the time and frequency domain perspective, the road condition significantly impacted the vehicle force on the bridge.

Figure 6a shows the variation curve of tire force of the middle wheel of the vehicle with driving time under different speed conditions when the road surface grade was B. Figure 6b shows the spectrum curve of tire force in the middle wheel of the vehicle at different driving speeds. Figure 7b shows the variation curve of the dynamic coefficient of the wheel at different driving speeds. Table 4 shows the value of the peak force.

**Table 4.** Wheel force data in different pavement grades.

| Vehicle Speed (km/h) | Road Grade | Middle-Wheel Acting Force (N) | Static Force Value | Dynamic Coefficient |
|---|---|---|---|---|
| 20 | grade A | 84,643 | 70,525 | 1.200 |
|  | grade B | 99,333 | 70,525 | 1.408 |
|  | grade C | 144,205 | 70,525 | 2.045 |
|  | grade D | 205,249 | 70,525 | 2.910 |
| 30 | grade A | 84,380 | 70,525 | 1.196 |
|  | grade B | 97,724 | 70,525 | 1.386 |
|  | grade C | 119,480 | 70,525 | 1.694 |
|  | grade D | 175,250 | 70,525 | 2.485 |
| 40 | grade A | 83,880 | 70,525 | 1.189 |
|  | grade B | 96,620 | 70,525 | 1.370 |
|  | grade C | 122,578 | 70,525 | 1.738 |
|  | grade D | 166,237 | 70,525 | 2.357 |
| 50 | grade A | 83,857 | 70,525 | 1.189 |
|  | grade B | 102,211 | 70,525 | 1.449 |
|  | grade C | 123,242 | 70,525 | 1.747 |
|  | grade D | 185,361 | 70,525 | 2.628 |

From Figure 6, it can be seen that the speed change had no evident impact on the force of the middle-particular tire at a particular driving position, but there was a specific fluctuation phenomenon, which may have been related to the speed changing the excitation characteristics. It can be seen from the Fourier spectrum that when the frequency was at 0.5 Hz~5.0 Hz, the Fourier spectrum curve changed toward the higher frequency, which was related to the increased speed increasing the frequency of the load.

### 4.2. Influence of Braking and Jumping on Vehicle Force

The braking and jumping test can reflect the bridge's resistance to a longitudinal and vertical impact in the bridge load test. Related research [21] noted that the braking force could be assumed as a ramp load, in which the braking force size increased linearly from zero to the maximum and remained constant until the vehicle stopped or left the bridge. The force of the vehicle on the bridge can assume a slope load, as shown in Equation (5):

$$F_{xt} = \begin{cases} -F_{xt\max}t/t_b, t < t_b \\ F_{xt\max}, t \geq t_b \end{cases} \tag{5}$$

$t_b$ is the braking force rise time, and 0.3 s was taken in this calculation; $F_{xt\max} = W \times \varphi$, $W$ is the vehicle dead weight, and $\varphi$ is the vehicle attachment coefficient which was taken as 0.7.

The literature [22] points out that the vehicle's movement after jumping over the springboard is relatively complex. Generally, in the bridge detection test, it can be considered that the vehicle is in a stationary state when it reaches the highest part of the springboard. During the initial calculation, the vertical constraint of the contact between the middle and rear wheels and the bridge deck is released, and it is assumed that the middle and rear wheels move freely. When they contact the bridge deck, the vertical displacement of the contact point is restrained, and then the vehicle moves freely. The constraint reaction of nodes is extracted.

Figure 8a shows the time travel curve of force in the vehicle after the tire and road contact when the vehicle jumps. It can be seen that the peak force of the middle wheel-tire reached $3 \times 10^5$ N, with a dynamic coefficient of 4.2. Figure 8b shows the mid-wheel load spectrum curve, and the peak appeared in the range of 0–3 Hz, which is consistent with the first third-order self-vibration frequency of the car body, with a peak at 27 Hz related to the vibration of the tire.

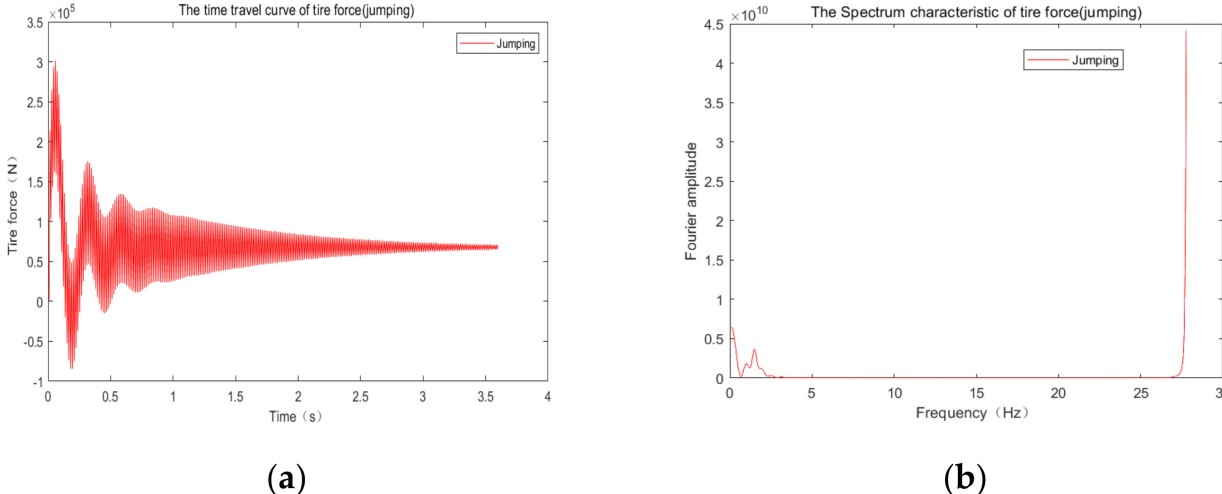

(a)                                          (b)

**Figure 8.** Time-history and spectrum characteristic curve of tire force upon jumping. (**a**) Time-history curve of tire force in vehicle; (**b**) Wheel force spectrum curve in vehicles.

## 5. Establishment and Verification of the Vehicle Axle Coupling Model

### 5.1. Establishment of the Bridge Coupling Model

The research object was a large urban municipal bridge, divided into four links, with a total length of 650 m. The layout of each joint span is successively: $3 \times 40$ m, $4 \times 40$ m, $5 \times 60$ m. Figure 9 shows the actual scene.

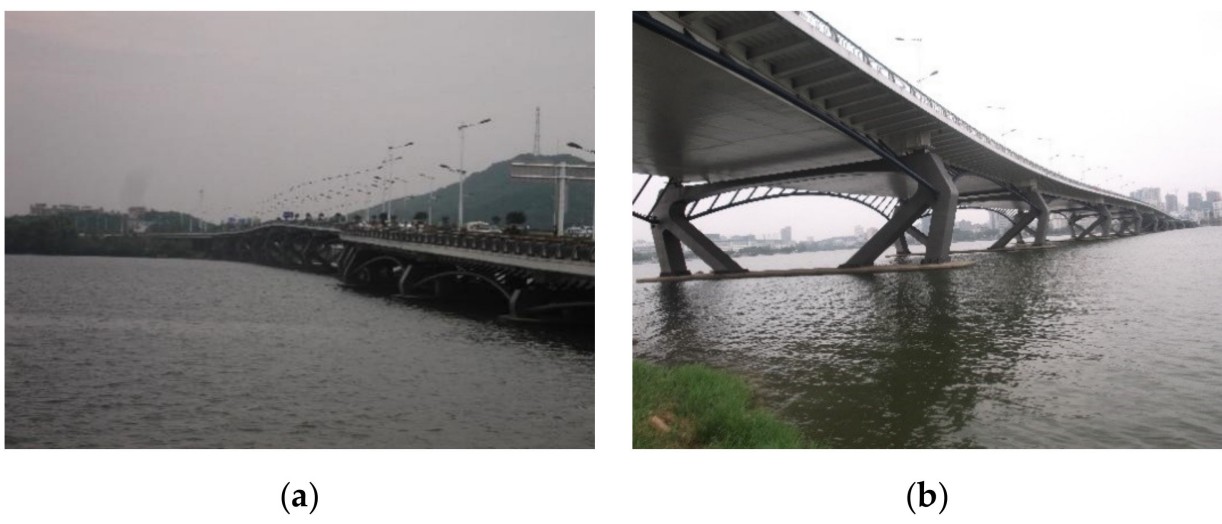

(a)                                          (b)

**Figure 9.** The bridge is built in real action. (**a**) Bridge; (**b**) pier of the bridge.

The bridge is a steel box girder bridge, and the steel plate thickness used was much less than the length and width, so the shell181 element was used to represent the reinforcement ribs of the bridge deck, bottom plate, box girder web, partition, deck and bottom plate. The influence of strengthening the rib on the bridge stiffness was considered in this model. The influence of stiffeners on bridge stiffness was considered, and the bridge bottom plate, transverse diaphragm access hole, and transverse diaphragm stiffeners were not considered. Figure 10 shows the bridge structure model. Figure 10b,c are schematic diagrams of sections of the bridge structure at the bearing position and the mid span position, respectively. Figure 10d is the partial schematic diagram of the bridge model at the middle bearing, in which longitudinal and transverse stiffening ribs were carefully considered.

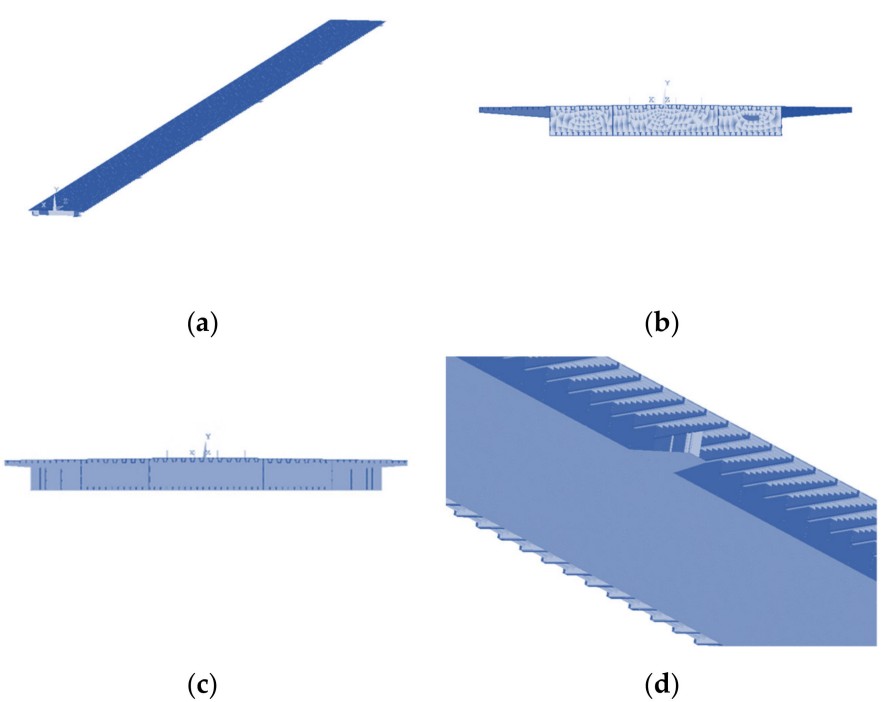

**Figure 10.** Bridge ANSYS FEM model. (**a**) Global model of the bridge; (**b**) Cross-section at mid span position; (**c**) Cross-section at bearing position; (**d**) Branch par.

The material used in the main structure was Q345D in China, and the total mass of the bridge span model was 3608 t. According to the constraints provided by the actual bearings, the degrees of freedom in the three directions of UX, UY, and UZ were constrained at the first row of bearings and the sixth row of bearings, while the vertical degrees of freedom of UY were constrained at other bearings. At the same time, only the UX of one bearing was constrained at the bearings from the second to the fifth. For the convenience of load application, the auxiliary beam without influence on the structural stiffness and mass was established on the model. Figure 11 shows the vehicle and the auxiliary beam model established based on the ANSYS program. The mass of the auxiliary beam was zero and the length was 1 m. The connection between the auxiliary beam and the bridge structure model was simulated by using the link180 element with rigid but zero mass. The connection between the auxiliary beam and link180 element was simply supported, and there was no node connection between adjacent auxiliary beams. The spacing between the link180 elements was 1 m. The auxiliary beam adopted a beam188 element, and the length of the element division was 0.1 m. The vehicle bridge coupling system calculated the dynamic response by coupling the displacement at the contact point between the vehicle and the bridge in each time step.

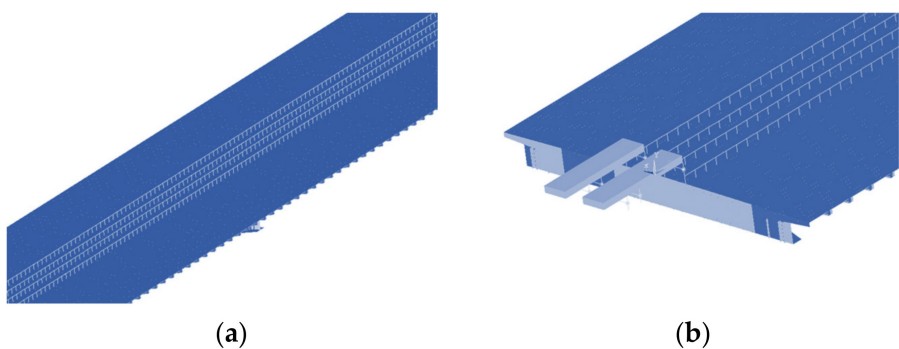

**Figure 11.** Model of local and auxiliary beams. (**a**) Local auxiliary beam; (**b**) Finite element model.

When the structure was meshed, the length of element was set to 1 m. Because the bridge was equipped with more stiffeners, the bridge structure was relatively complex, and the number of model elements and nodes was large. The vehicle bridge coupling finite element model had a total of 169,554 elements and 152,904 nodes.

During the calculation, the time step was set as the time for the vehicle to travel 0.1 m, so the maximum time step was $\Delta t = 0.1/(20/3.6) = 0.018$ s, and the eighth natural vibration period of the bridge was $T = 0.202$ s, so $\Delta t/T = 0.08 \leq 0.1$, which met the requirements of structural calculation. The method of Newmark-β was used for numerical integration.

### 5.2. Model Test Validation

The bridge structure produced vibrations caused by vehicle or environmental excitation. The response of the bridge was measured by vibration pickup, and the dynamic DHDAS (5922-1394) was used for real-time measurement and analysis. The actual scene is shown in Figure 12.

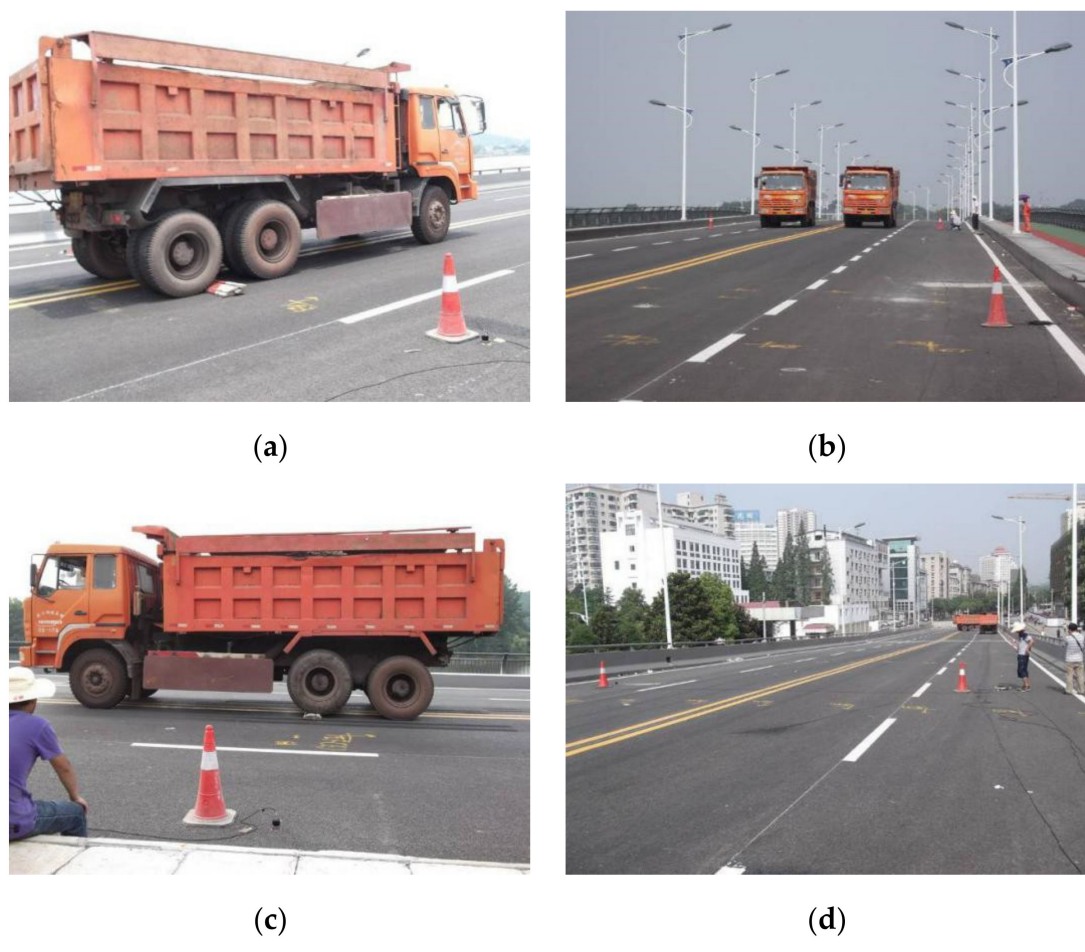

**Figure 12.** Bridge load test. (**a**) Jumping test; (**b**) Sporting test; (**c**) Braking test; (**d**) Pulsation test.

The natural characteristics of the conventional bridge FE model were analyzed, and the structure's first eight natural vibration frequencies were obtained. Table 5 describes the bridge span structure's natural frequency, period, and vibration mode.

Compared with the pulsation test, the calculated frequency was higher, which was related to the fact that the existing maintenance hole and welding reduced the bridge stiffness, but the error was controlled by 10%. The established FE bridge model met the calculation requirements. Figure 13 shows the vibration mode diagram of the first four modes.

**Table 5.** The self-vibration frequency of the bridge.

| Order | Natural Frequency Of Vibration (Hz) | Natural Vibration Period (s) | Measured Frequency (Hz) | Error ((Self-Vibration Frequency-Measured) /Measured) | Vibration Description |
|---|---|---|---|---|---|
| 1 | 1.72 | 0.582 | 1.65 | 4.24% | First-order vertical symmetric bending vibration |
| 2 | 1.82 | 0.550 | 1.75 | 3.94% | First-order vertical antisymmetric bending vibration |
| 3 | 2.22 | 0.449 | 2.15 | 3.49% | Second-order vertical symmetric bending vibration |
| 4 | 2.57 | 0.388 | 2.45 | 5.14% | Third-order vertical antisymmetric bending vibration |
| 5 | 2.92 | 0.343 | 2.76 | 5.65% | Fourth-order vertical antisymmetric bending vibration |
| 6 | 2.98 | 0.336 | 2.78 | 7.01% | Third-order vertical symmetric bending vibration |
| 7 | 4.94 | 0.203 | 4.51 | 9.49% | First-order bending and torsion are coupled to the vibration |
| 8 | 4.94 | 0.202 | 4.62 | 6.90% | Second-order twisted-coupled vibration |

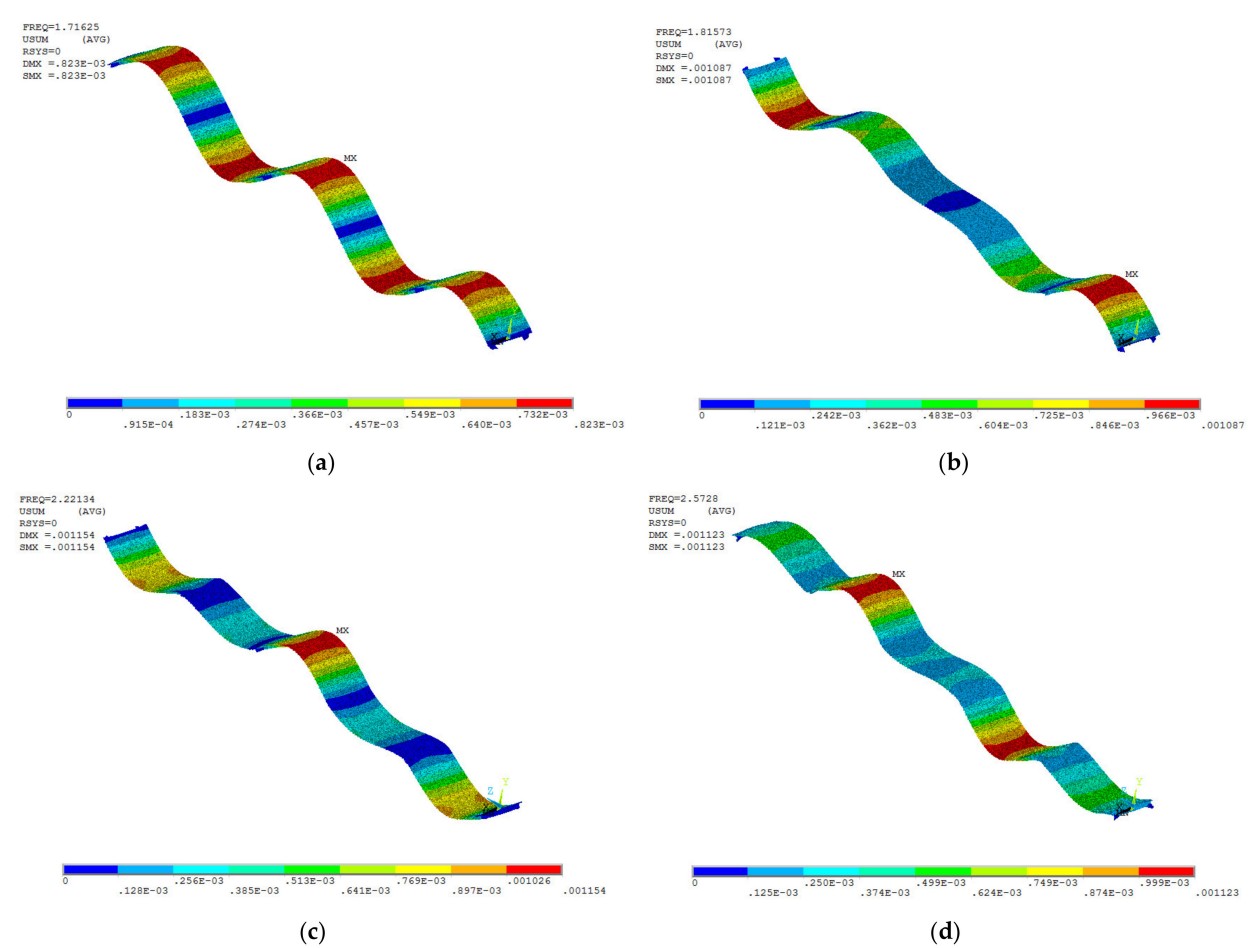

**Figure 13.** Modes of the bridge. (**a**) The first order of bridge; (**b**) The second order of bridge; (**c**) The third order of bridge; (**d**) The fourth order of bridge.

*5.3. Comparison of the Bridge Dynamic Response Calculation and the Test*

Figure 14a–h shows the time-history and spectrum comparison curves of the calculated and bridge test acceleration at the middle of the second span at 20 km/h and 40 km/h.

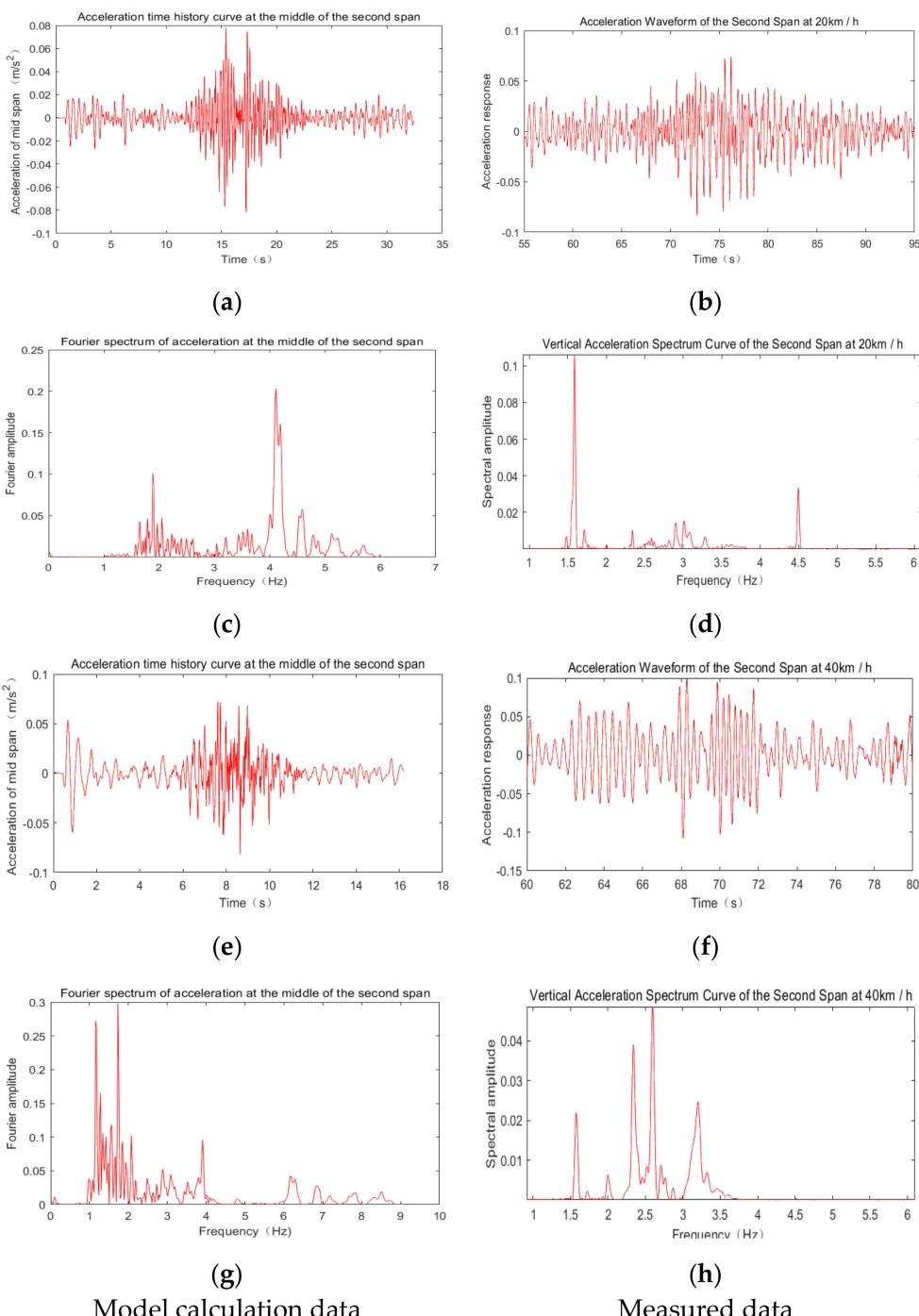

Model calculation data                                    Measured data

**Figure 14.** Comparison of the acceleration measurement and calculation. (**a**) Calculated acceleration time-history of 20 km/h; (**b**) Test acceleration time-history of 20 km/h; (**c**) Calculated acceleration spectrum of 20 km/h; (**d**) Test acceleration spectrum of 20 km/h; (**e**) Calculated acceleration time-history of 40 km/h; (**f**) Test acceleration time-history of 40 km/h; (**g**) Calculated acceleration spectrum of 40 km/h; (**h**) Test acceleration spectrum of 40 km/h.

Table 6 shows the peak acceleration. The data show that the error was controlled by 20%. Because there were many factors affecting the measured peak acceleration, which

was affected by the actual road roughness, vehicle parameters, external excitation, and other factors, the calculation met the requirements from the perspective of the calculation verification angle. From the time-history waveform curve, the test waveform curve was smoother than the calculation curve, related to the discrete moving force used in the calculation process. It should be a continuous moving force. The calculated and measured peak acceleration roughly appeared when the vehicle passed through the measuring point. The attenuation speed of the calculated acceleration was faster than the measured one, which may be related to the insufficient consideration of the influence of the carriageway slab when the model was established. The measured data and calculated data of jumping also show the same problem. The calculation was conservative. The measured and calculated data were consistent. The measured and calculated peak values appeared from the spectrum curve appear between 1 and 6 Hz, and the amplitude at the corresponding frequency was the same, but the calculated high-frequency band was more affluent. The above description shows that the calculation method can meet the needs of practical engineering calculations.

**Table 6.** Comparison between the measurement and calculation of peak acceleration (m/s$^2$).

| Location / Test Type | The Measurement of the Second Span (A) | The Calculation of the Second Span (B) | Error \|(A−B)/A\| | The Measurement of the First Span (A) | The Calculation of the First Span (B) | Error \|(A−B)/A\| |
|---|---|---|---|---|---|---|
| 20 km/h | 0.076 | 0.078 | 2.63% | 0.073 | 0.075 | 2.74% |
| 30 km/h | 0.124 | 0.101 | 18.55% | 0.062 | 0.069 | 11.29% |
| 40 km/h | 0.096 | 0.084 | 12.50% | 0.078 | 0.089 | 14.10% |
| Jumping | 0.122 | 0.142 | 16.39% | | | |

## 6. Dynamic Response of the Bridge Structure

### 6.1. Influence of the Vehicle Driving Speed

It was assumed that the vehicle passed the bridge deck at the speed of 20 km/h, 30 km/h, 40 km/h, and 50 km/h, respectively, and the pavement grade was B. Two vehicles drove side by side along the center line of the bridge at a uniform speed, and the corresponding midspan displacement time-history was calculated. Figure 15 shows the time-history waveform of vertical displacement in the middle of the second span, and Figure 16 shows the acceleration spectrum at the corresponding positions at different vehicle speeds.

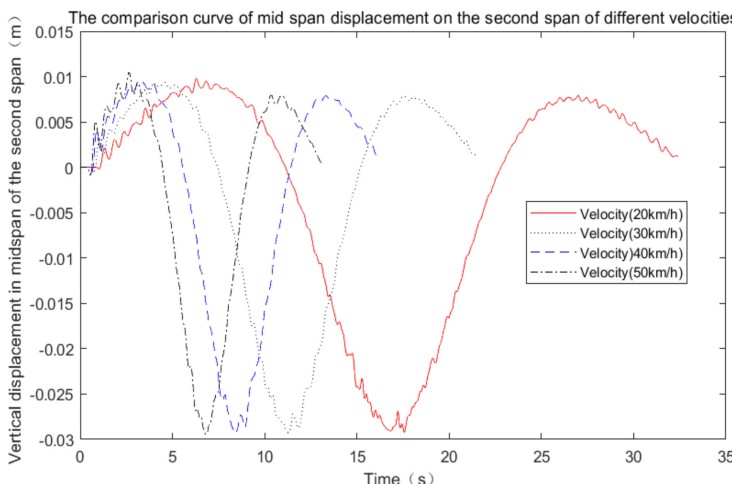

**Figure 15.** Waveform diagram of the vertical displacement in the second span.

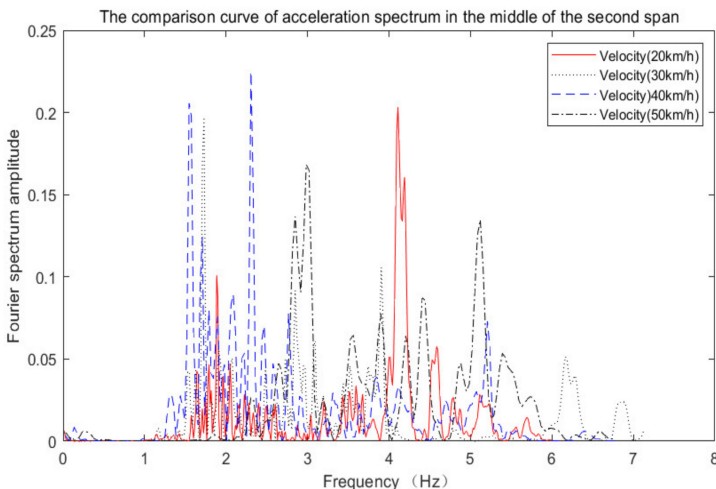

**Figure 16.** The vertical acceleration spectrum curves across the second span.

Table 7 shows the displacement peak values at the corresponding positions. Figure 17 shows the displacement peak values under different vehicle speeds. From the perspective of peak displacement, the speed had little effect on the displacement response, with slight fluctuations.

**Table 7.** Peak displacement at a corresponding position at different vehicle speeds (mm).

| Position \ Speed | 20 km/h | 30 km/h | 40 km/h | 50 km/h | Maximum Static Displacement |
|---|---|---|---|---|---|
| The first span | 34.83 | 34.45 | 34.44 | 35.71 | 26.59 |
| The second pan | 29.25 | 29.26 | 29.30 | 29.35 | 21.28 |
| The third span | 29.06 | 28.57 | 28.69 | 28.98 | 20.99 |

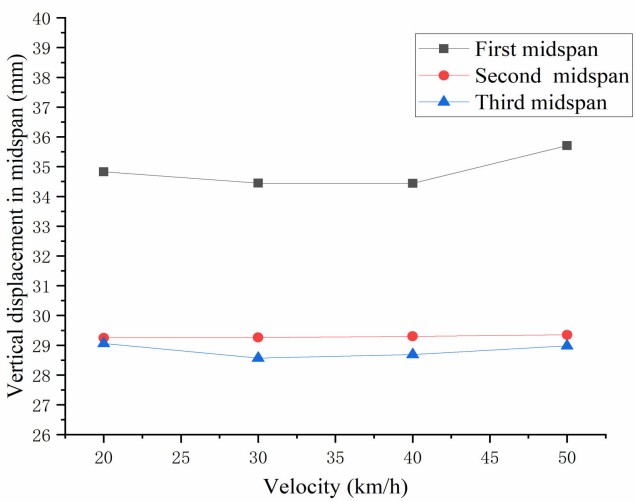

**Figure 17.** Comparison of peak displacement at different vehicle driving speeds.

The deflection of the second and third spans was the same, consistent with the law of maximum static displacement. From the time-history waveform, it can be seen that the peak position of -mid-span deflection appeared when the vehicle travelled to the measuring point.

From the acceleration spectrum curve, it can be seen that the change of speed had no impact on the spectrum in a specific direction, and the peak value appeared between 1~7 Hz. The results show that although the speed changed the characteristics of the vehicle tire force, it had no significant impact on the bridge due to the complex influence of bridge

natural frequency characteristics and vehicle bridge coupling. The first span and the third span also show the above characteristics. Similar characteristics were shown under other pavement grade conditions.

### 6.2. Influence of Deck Roughness on the Dynamic Response of the Bridge Span Structure

In the national standard, the vehicle travels on the bridge decks of pavement grades A, B, C, and D with a speed 40 km/h. It shows the waveform of vertical displacement in the second midspan with different pavement grades in Figure 18. Figure 19 shows the corresponding acceleration spectrum.

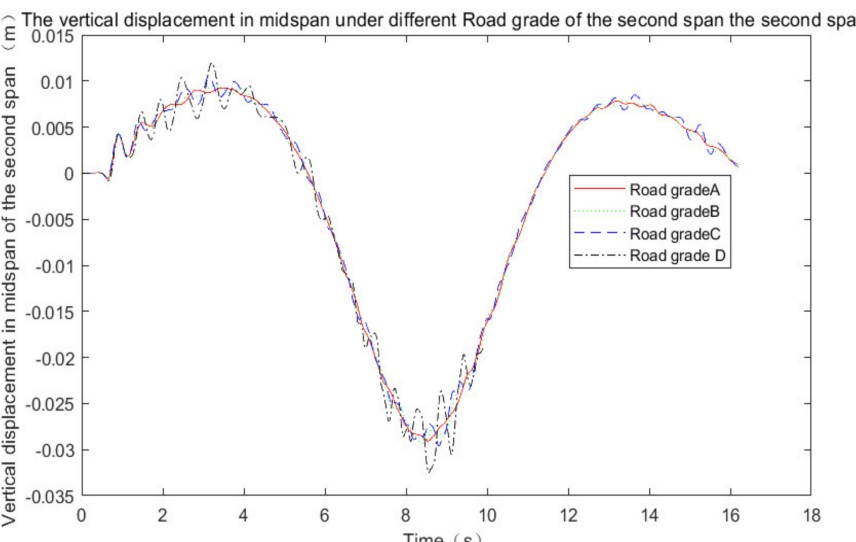

**Figure 18.** Time-history waveform of different pavement grades.

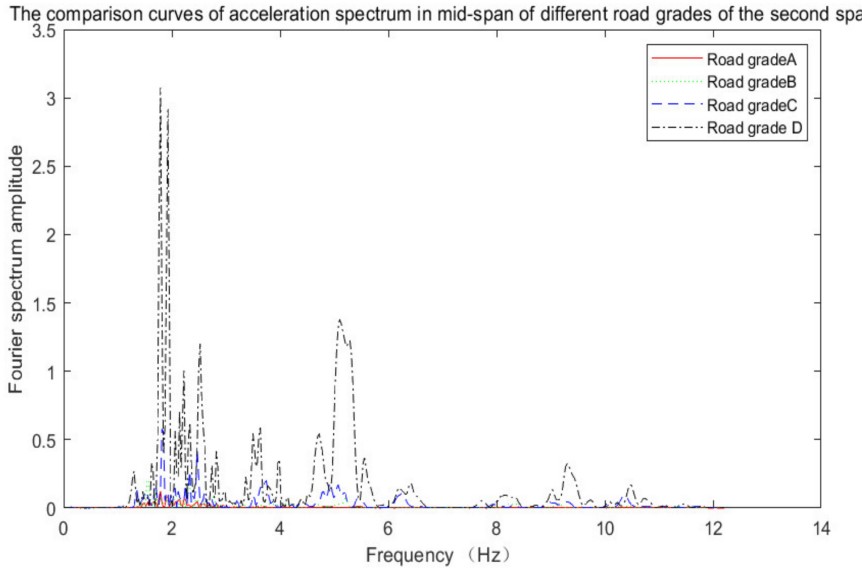

**Figure 19.** Comparison of vertical acceleration spectrum curves of different pavement grades.

Table 8 lists the displacement peak values at the corresponding positions under different pavement grades. Figure 20 shows the comparison diagram of displacement peak values under different pavement grades. From the perspective of peak displacement, the pavement grade had no significant impact on the bridge. With the deterioration of pavement conditions, the dynamic response of the bridge tended to increase.

**Table 8.** Peak displacement at corresponding positions under different pavement grades (mm).

| Grade / Location | Grade A | Grade B | Grade C | Grade D | Maximum Static Position Move |
|---|---|---|---|---|---|
| The first span | 34.33 | 34.44 | 36.58 | 39.65 | 26.59 |
| The second span | 29.09 | 29.30 | 29.63 | 32.33 | 21.28 |
| The third span | 28.80 | 28.69 | 29.36 | 30.58 | 20.99 |

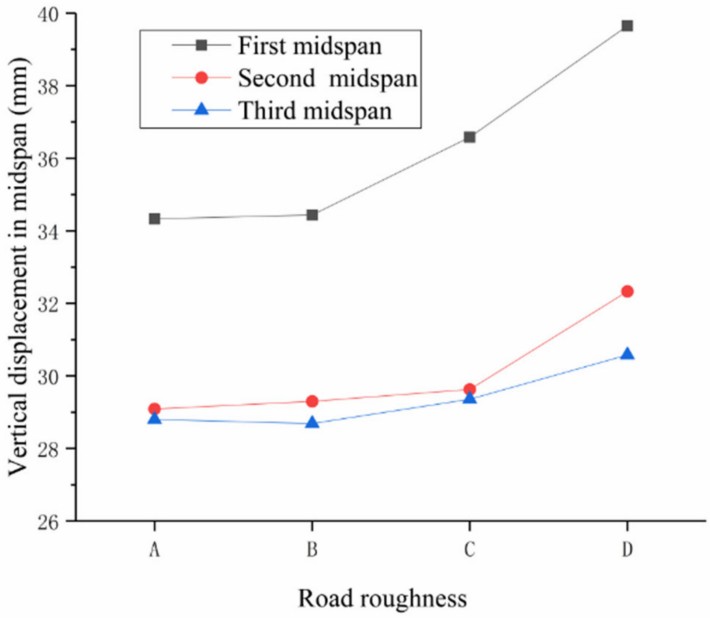

**Figure 20.** Comparison of displacement peak values of different road grades.

According to the corresponding time-history waveform, the fluctuation of the waveform increased significantly with the deterioration of the pavement grade. From the corresponding acceleration spectrum curve, the deterioration of the pavement grade and the frequency at the peak of the spectrum curve did not change significantly. However, the amplitude increased significantly, which was related to the different pavement roughness coefficients but the consistent law of the spectrum curve. The positions in the middle of the first and the third spans showed the same characteristics. Similar characteristics were shown at other speeds.

### 6.3. Influence of the Vehicle Eccentric Load on the Dynamic Response of Bridge Span Structure

Because of the fact that two vehicles drove side by side 4.0 m away from the center, the displacement time histories of the side near the vehicle load bridge flange and the side away from the vehicle load bridge flange at the mid-span position were calculated, respectively. The vehicle speed was 40 km/h, and the pavement grade was B. Figure 21 shows the displacement response waveform of the bridge's second span. Figure 22 shows the corresponding spectrum curve.

Table 9 lists the peak displacement under an eccentric load. The table shows that the peak displacement at the flange near the vehicle load was significantly greater than that without an eccentric load, while the peak displacement at the flange away from the load was significantly lower than that without an eccentric load. The spectrum characteristics did not change significantly from the spectrum curve. However, the amplitude of the frequency above 5 Hz significantly increased, which also illustrates that the effect of the load on the bridge was more concentrated in the frequency band above 5 Hz, and the positions of the first and third span showed the same characteristics. Other speeds and unevenness also showed similar characteristics.

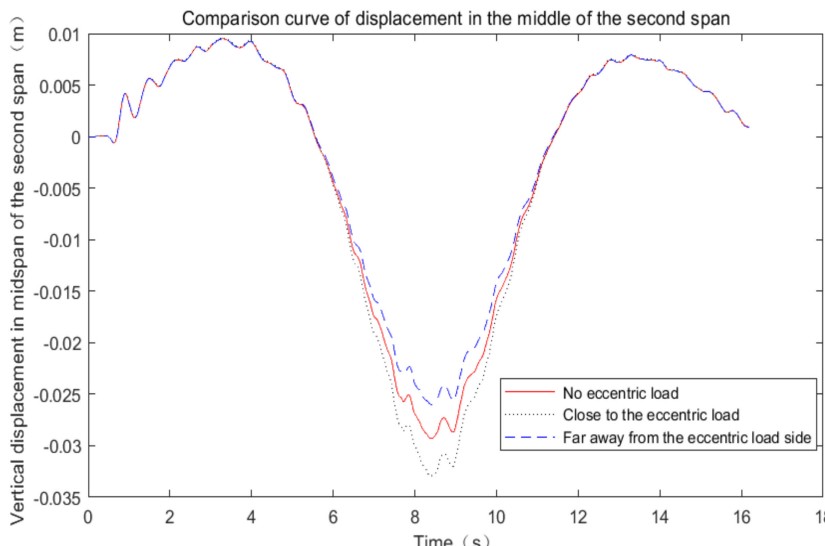

**Figure 21.** Comparison diagram of the displacement response waveform under an eccentric load.

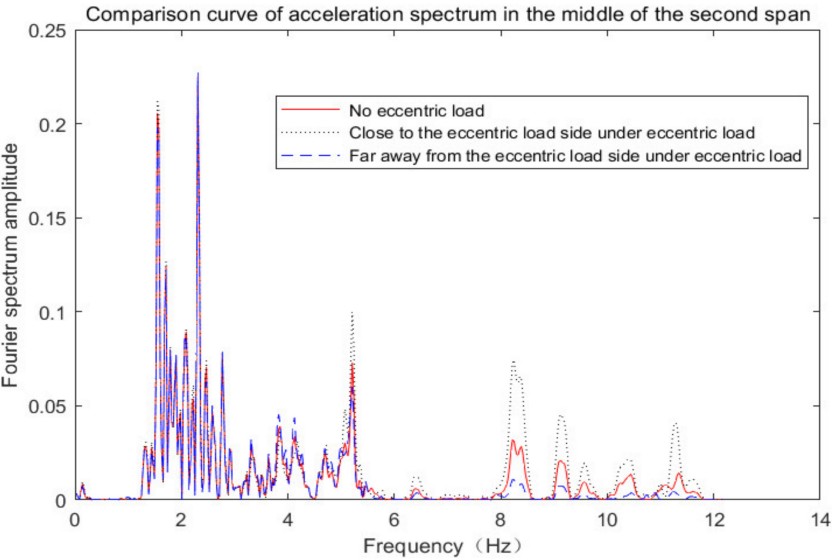

**Figure 22.** Comparison diagram of an acceleration spectrum curve under an eccentric load.

**Table 9.** Peak displacement under an eccentric load (mm).

| Condition<br>Location | Without Eccentric Load | Close to the Load | Away from the Load |
|---|---|---|---|
| The first span | 34.44 | 38.20 | 31.25 |
| The second span | 29.30 | 32.96 | 25.99 |
| The third span | 28.69 | 32.42 | 25.39 |

### 6.4. Influence of Vehicle Braking

The speed before braking was 20 km/h and 40 km/h, respectively, and the braking position was at the first mid-span. Figure 23 shows the displacement response waveform at the braking position under two braking speeds, and Figure 24 shows the corresponding spectrum curve. Different braking speeds did not affect vertical displacement. From the spectrum diagram, there were more high-frequency parts of the acceleration spectrum diagram at higher speeds, and the braking at other positions also had the same characteristics.

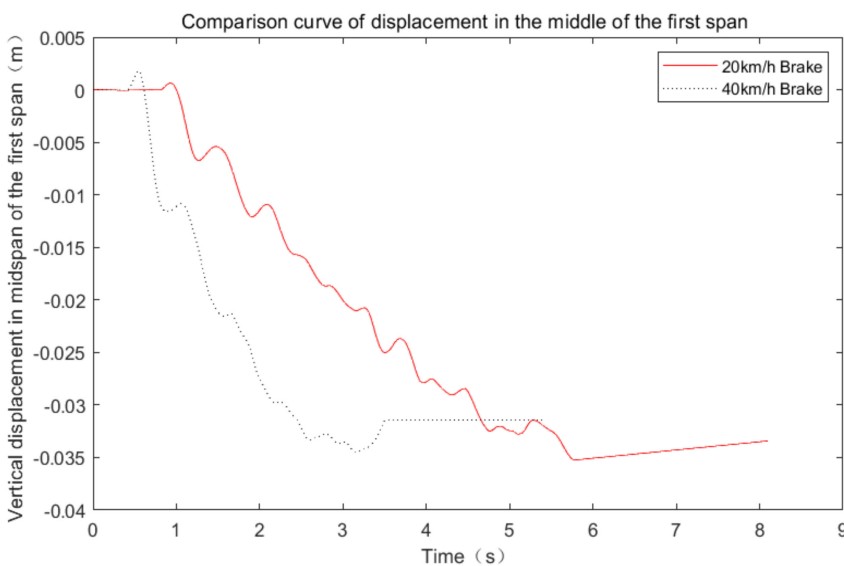

**Figure 23.** Displacement response waveform at different brake speeds.

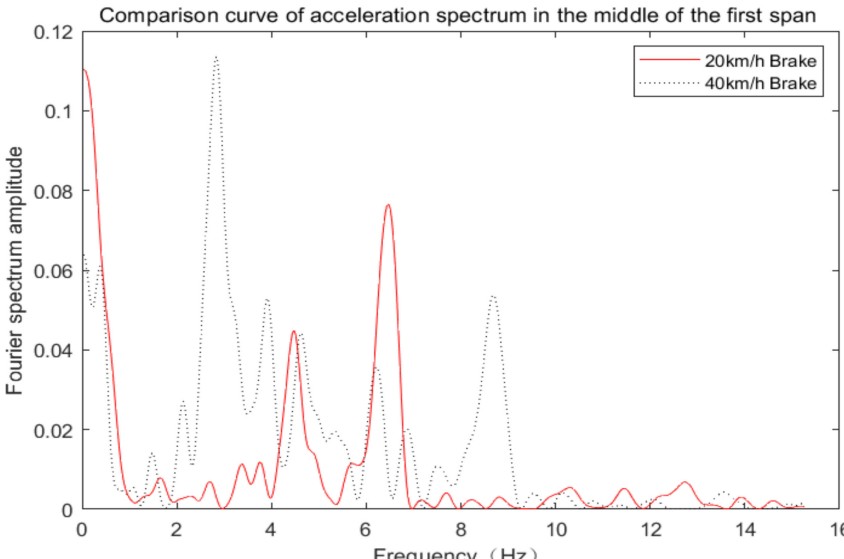

**Figure 24.** Acceleration spectrum curve at brake positions at different brake speeds.

## 7. Conclusions

The ANSYS FE model was established according to the original design data. The measured values were used to prove the model's accuracy, and the accuracy met the requirements of the engineering calculation. The unevenness excitation was applied to the vehicle, the corresponding vehicle force was extracted, and the vehicle force was applied to the bridge structure in moving force. The dynamic response was calculated in the coupling system by successively coupling the displacement at the contact point in each time step. The effects of different vehicle speeds, road conditions, whether the vehicle was biased or not, vehicle braking on the displacement, and the acceleration were analyzed:

(1)　The vehicle speed had no significant influence on the displacement time-history and the force of the middle wheel of the vehicle at a specific driving position. A particular fluctuation phenomenon may be related to the speed's change of the excitation characteristics. With the speed increase, the spectrum curve did not change in a specific direction, and the peak value appeared between 1~7 Hz, although the speed changed the influence of the force of the vehicle tire. However, due to the influence of the natu-

ral frequency characteristics of the bridge and the complex vehicle bridge coupling, we did not find the law of the influence of speed change on the spectrum characteristics.

(2)    The pavement grade significantly influenced the bridge's displacement time-history and acceleration spectrum. With the deterioration of road conditions, the peak force of the vehicles gradually increased, the amplitude of the frequency spectrum also gradually increased, the peak displacement of the bridge increased significantly, the fluctuation of vertical displacement waveform at the midspan position increased significantly, and the frequency at the peak of vertical acceleration spectrum curve at the midspan position did not change significantly. The corresponding amplitude of the frequency band above 5 Hz increased significantly.

(3)    In the case of an eccentric vehicle load, the peak displacement of the flange near the vehicle position was significantly greater than that without an eccentric load, while the peak displacement of the flange away from the vehicle position was less than that without an eccentric load. The spectrum characteristics of the mid-span vertical acceleration spectrum curve did not change significantly, but the frequency amplitude above 5 Hz increased significantly.

(4)    Different braking speeds had no impact on the vertical displacement. The acceleration spectrum pattern was distributed more at high speeds in the high-frequency bands.

**Author Contributions:** Conceptualization, B.W.; methodology, Y.J.; software, H.Z.; validation, S.W., Z.J. and J.Y.; formal analysis, B.W.; resources, B.W.; data curation, B.W.; writing—original draft preparation, B.W.; writing—review and editing, J.Y.; visualization, S.W.; supervision, Baitian Wang; project administration, B.W.; funding acquisition, B.W. All authors have read and agreed to the published version of the manuscript.

**Funding:** This research was funded by the 14th five year plan project of Educational Science in Henan Province (2021YB0474). The research and practice project of new engineering in Henan Province (2020jglx091). The innovation training project of college students in Henan Province (S20214003004). Henan Academy of Civil Architecture Science and Technology Special Project (202103). General project of Humanities and Social Sciences in Colleges and universities of Henan Province (2023-zdjh-013). The higher education research project of Henan Higher Education Society (2021sxhlx179).

**Informed Consent Statement:** Not applicable.

**Data Availability Statement:** The data were generated from experiments and are available from the corresponding author upon request.

**Conflicts of Interest:** The authors declare no conflict of interest.

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
