# Peer review of "Research on the Dynamic Response of a Continuous Steel Box Girder Bridge Based on the ANSYS Platform"

_sustainability, doi:10.3390/su141710638_

Round 1

Reviewer 1 Report

The provided manuscript “Research on dynamic response of continuous steel box girder bridge based on ANSYS platform” delivers well-rounded research on the analysis of the vehicle effects on the bridge response and how the whole dynamic response is changed under different conditions. The theoretical evaluation is supported also by experimental measurements. Generally, the manuscript is written in a clear manner, but some details need to be provided and some reconsideration regarding the modelling conditions and boundaries should be revisited for the manuscript to be of higher value and provide some more novelty in their research field.

I would recommend the authors to explain more about the actually implemented theoretical models that were used within the ANSYS mainframe. It would be good to provide as Supplementary some basics about what modules/routines were used for their calculations. The authors should also try to provide also a deeper motivation in what is missing from others research by giving some clear fallacies of previous research or missing considerations of others.

Please give reasons and/or references for selected parameters provided in Table 1. Why did you choose these parameters? Maybe state the origin of the data to correlate to real-world values to give a clear reason why these values were chosen (maybe give an example model of the truck etc.).

Generally, I have a concern regarding the actual roughness effect on the bridge force. The calculations seem to approach the phenomenon as a single point effect. However, the actual dampening effect is caused also by the elasticity of the tire and of the tire rim. Furthermore, the weight distribution is not clearly given or described. The tire is not a single point object and distributes the weight along its cut-off curvature that is pressed to the road. The flatness of the tire will be in the cm range so well above the roughness change of the road. How do the authors correlate their answer without assuming such effects of the tire and its flat area distribution of the forces that effectively damp the actual change in height of the vehicle? I believe this is an important additional parameter that needs to be considered in order to get the real resonance change of the vehicle caused by road roughness. Note, in this case also the tire elasticity, radius, pressure and weight distribution on them has to be well determined. In this case I believe a separate model of the entire tire system is needed in order to consider all these effects that will have a high impact on the data presented in sections 3 and 4. With such a model the vehicle response will be more correctly calculated and provide a more smoother frequency response as those measured with real tests from section 5.

For the driving speed effects: Did the authors considered the effect of the road roughness on the reduction of the speed of the vehicle. If so, please explain how this was done as it is not clear from the currently provided manuscript. If this was not performed, then the calculation must be performed with drag effect of the pavement on the speed as this leads to a deacceleration effect that is a common feature of vehicles on pavements.

For the braking, please provide the breaking parameters used. In this regard did the authors consider the “grabbing” of the tire material on the pavement or did they consider only as a single point deacceleration with only a nominal force opposite to the driving direction? This needs to be more clearly laid out in the manuscript and also evaluated correctly to assume the effect of tire partially rotating with the breaking (the real tire does not simply stop rotating with breaking, but also does not role proportionally with the path taken by the vehicle). I would ask the authors to give some insight into this phenomenon.

Some other general comments regarding text:

Please define all acronym that firstly appear in the manuscript, example: Line 45: define what is MBS, Line 50: define what is GHW, Line 59: define what is FEM

Table 1: change M4 definition to Body mass (without y at the end)

Author Response

Dear editors and reviewers.

      Thank you very much for your constructive comments on the article. These comments are distributed in all parts of the article. These comments are innovative in combination with the article. The opinion makes an in-depth analysis of the problems existing in the analysis and calculation of the article, which is conducive to improving the quality of the article.Combined with the comments, the author revised the article and responded to the questions raised by the reviewers one by one. Please see the attachment for more details.

Reviewer 2 Report

Dear Authors,

MAJOR REMARKS

1. In my opinion, the subject of the proposed paper does not refer to the Aims & Scope of the Journal Sustainability.

2. The format and style of the paper do not correspond to the template of the Journal.

3. The paper is difficult to read due to the grammar/style as well as the form of the presentation and discussion of the results. Many figures (no. 4a-d, 5a-d, 6a-b, 10a-d,11a-b,13a-d, 19, 22) are completely unable to read. The discussions of the obtained results and applied models are also poor or absent.

4. What about the influence of vehicle speed on the resonance phenomenon?

5. References should be revised. I.e there are missing references in lines  50, 51.

6. What is the novelty of the paper?

7. Equation 2 and Lines 82-88 – It should be revised (not clear and mistakes).

8. Line 89 – pavement grades A,B,C - not discussed in any way before.

9. Figure 7 seems to be wrong – the vertical axis is dynamic coefficient while in the caption is “peak force”. What is the dynamic coefficient?

10. Line 167 – What means “the vehicle dead weight” and how it was calculated?

11. Figures 10 and 11 – Figures are really difficult to read and understand. The form of the presentation should be changed and improved. What about the influence of the FE mesh on convergence of the numerical solution?

12. Tables 5 and 6 – I propose to give an error in %.

13. Table 6 is difficult to understand (description).

Minor remarks

1. Abstract Line 17: What language was used?

2. Abstract Line 19 “ correctness” – I propose accuracy

3. Line 37: “Timoshenhenko” – it is a mistake.

4. Line 149 – hz should be corrected.

5. Figure 20- the color of the triangle in the legend is wrong.

Kind Regards

Author Response

(The authors gave the same response as above.)

Round 2

Reviewer 1 Report

The authors have answered all my questions and given a good response. Despite some assumptions being maybe too simplistic to envelope the whole interaction of the vehicles with the bridge structure, I believe this work is still an important contribution to the engineering fields related to bridge construction and design. As such I suggest the paper to be accepted for publication. I look forward to reading more detailed work from the authors on this research topic.

Author Response

Dear editors and reviewers.

     Thank you very much for your constructive comments on the article.We appreciate for Editors/Reviewers’ warm work earnestly.We carefully checked and revised the English format and grammar of the manuscript.  Once again, thank you very much for your comments and suggestions.

Reviewer 2 Report

Dear Authors,

Thank you for your detailed response for my remarks, however I still have following remarks:

1. Line 77 „The Chinese national standard (GB /T7031-86)’ Vehicle Vibration input -Representation of Road Roughness’” – The reference should be given

2. Line 108-109 – please correct the formatting of the text and title of subsection 3.2

3. figures 3b, 3c, and 3d still are blurred and without description.

4. Figure 4 –  Why the plots are distorted by black/grey lines? What does it mean? The font size in the legend should be larger. Figure 4c is difficult to understand – This example should be clarified in the text. I propose also to put Table 2 before Figure 4.

5. The whole manuscript should be revised concerning missing or too many spaces, dots, etc. I also recommend the revision of the manuscript by a native English speaker.

6. Line 139 – Is it the same orientation of the local coordinate system as it is given in Figures 3 and 4?

7. Line 160 and Figure 7 and comment to Authors response: 8.Response to comment: Figure 7 seems to be wrong – the vertical axis is dynamic coefficient while in the caption is “peak force”. What is the dynamic coefficient?

Response:The general value of dynamic coefficient is the ratio of the maximum dynamic effect of the structure to the corresponding static effect. For the convenience of expression, the ratio of peak force to static force is used as the analysis object – The definition of the dynamic coefficient should be given and the description in the text and figure captions should be corrected.

8. Line 191 – missing unit after “5x60”

9. Figure 10 – pictures are still difficult to understand and must be improved. There are different colors without legend and descriptions, figures are blurred. What is at the upper-center part of Figure 10d (the place in which purple and light-blue colors change their shape)?

10. Figure 13 - The font size in the legends should be larger. I recommend improving the plots by removing gray distortions.

11. Figure 20 – The figure is blurred and must be improved.

 Kind Regards,

Author Response

Dear editors and reviewers.

      Thank you very much for your constructive comments on the article. According to the comments of the reviewers, we carefully revised the contents of the manuscript and English grammar.Please see the attachment.We tried our best to improve the manuscript and made some changes in the manuscript.      

     We appreciate for Editors/Reviewers’ warm work earnestly, and hope that the correction will meet with approval.
    Once again, thank you very much for your comments and suggestions

Round 3

Reviewer 2 Report

Dear Authors,

Remarks:

The quality of FEM plots (Figure 4) should be improved.

Kind Regards,

Author Response

Dear editors and reviewers.
    Thank you very much for your constructive comments on the article. We have modified the problem of Figure 4. We believe that the purpose of the vehicle modal analysis is to verify the accuracy of the vehicle model, and the high-order frequency has little effect on the dynamic response of the bridge. Therefore, for Fig. 4, in order to make the article more concise, we retain the first two modal cloud diagrams and delete Fig. 4C and Fig. 4D.
    We appreciate for Editors/Reviewers’ warm work earnestly
    Once again, thank you very much for your comments and suggestions.